# The Effect of Dietary Plant-Derived Omega 3 Fatty Acids on the Reproductive Performance and Gastrointestinal Health of Female Rabbits

**DOI:** 10.3390/vetsci11100457

**Published:** 2024-10-01

**Authors:** Alda Quattrone, Rafik Belabbas, Nour Elhouda Fehri, Stella Agradi, Silvia Michela Mazzola, Olimpia Barbato, Alessandro Dal Bosco, Simona Mattioli, Sebastiana Failla, El-Sayed M. Abdel-Kafy, Bayrem Jemmali, Imène Ben Salem, Maria Teresa Mandara, Giuseppe Giglia, Michel Colin, Mathieu Guillevic, Gerald Muça, Majlind Sulçe, Marta Castrica, Bengü Bilgiç, Maria Laura Marongiu, Gabriele Brecchia, Giulio Curone, Laura Menchetti

**Affiliations:** 1Department of Veterinary Medicine and Animal Sciences, University of Milan, Via dell’Università 6, 26900 Lodi, Italy; alda.quattrone@unimi.it (A.Q.); nour.fehri@unimi.it (N.E.F.); gabriele.brecchia@unimi.it (G.B.); giulio.curone@unimi.it (G.C.); 2Laboratory of Research “Health and Animal Productions”, Higher National Veterinary School, Road Issad 26 Abes, Oued Smar, Algiers 16200, Algeria; r.belabbas@ensv.dz; 3Department of Veterinary Sciences, University of Torino, Largo Paolo Braccini 2, 10095 Grugliasco, Italy; 4Department of Veterinary Medicine, University of Perugia, Via San Costanzo 4, 06126 Perugia, Italy; olimpia.barbato@unipg.it (O.B.); maria.mandara@unipg.it (M.T.M.);; 5Department of Agricultural, Food and Environmental Science, University of Perugia, Borgo XX Giugno 74, 06124 Perugia, Italy; alessandro.dalbosco@unipg.it (A.D.B.); simona.mattioli@unipg.it (S.M.); 6Consiglio per la Ricerca in Agricoltura e l’Analisi Dell’Economia Agraria (CREA), Centro di Ricerca Zootecnia e Acquacoltura, Research Centre for Animal Production and Aquaculture, Via Salaria 31, 00015 Rome, Italy; sebastiana.failla@crea.gov.it; 7Animal Production Research Institute (APRI), Agriculture Research Center (ARC), Dokki, Giza 12651, Egypt; sayedabdkaffy@yahoo.com; 8LR13AGR02, Higher School of Agriculture, University of Carthage, Mateur 7030, Tunisia; 9Département des Productions Animales, Service de Zootechnie et Economie Agricole Ecole Nationale de Médecine Vétérinaire, Université de la Manouba, Sidi Thabet 2020, Tunisia; bensalemimen@yahoo.fr; 10COPRI Sarl, Coat Izella 2, 29830 Ploudalmezeau, France; 11Valorex, La Messayais, 35210 Combourtillé, France; m.guillevic@valorex.com; 12Faculty of Veterinary Medicine, Agricultural University of Tirana, Kodër Kamëz, 1029 Tirana, Albania; gmuca@ubt.edu.al (G.M.); msulce@ubt.edu.al (M.S.); 13Department of Comparative Biomedicine and Food Science, University of Padova, Agripolis, Viale dell’Università 16, 35020 Legnaro, Italy; marta.castrica@unipd.it; 14Department of Internal Medicine, Faculty of Veterinary Medicine, Istanbul University, Cerrahpasa, Avcilar, 34093 Istanbul, Turkey; bengu.bilgic@iuc.edu.tr; 15Department of Veterinary Medicine, University of Sassari, Via Vienna 2, 07100 Sassari, Italy; marongiu@uniss.it; 16School of Biosciences and Veterinary Medicine, University of Camerino, Via Circonvallazione 93/95, 62024 Matelica, Italy; laura.menchetti@unicam.it

**Keywords:** algae, linseed, fertility, n-3 PUFA, nutraceutical, rabbit does, gastrointestinal histology, *Padina pavonica*

## Abstract

**Simple Summary:**

Different sources of omega-3 (n-3) fatty acids can influence various aspects of female reproduction. This study evaluated the effects of dietary n-3 inclusion with extruded linseed and algae *Padina pavonica* extract on the reproductive performance and milk production of nulliparous female rabbits. In this study, the dietary inclusion of extruded linseed and alga *Padina pavonica* extract did not alter the feed intake, body weight, or gastrointestinal health of the rabbits, suggesting good tolerability and the absence of adverse effects. No significant differences were observed in the reproductive parameters such as litter size, litter weight, and milk production. Our study revealed a significant decrease in perinatal and pre-weaning mortality rates among rabbit kits born to mothers receiving n-3 integrated diets. This finding likely results from the transfer of n-3 from mother to offspring during both pregnancy and lactation, potentially strengthening the litters’ immune defenses. While further studies are needed to elucidate the underlying mechanisms of action, dietary integration with extruded linseed and alga *Padina pavonica* extract shows promise as a strategy to physiologically enhance female rabbit reproductive performance and improve the survival of the litter, all in accordance with animal welfare principles.

**Abstract:**

This study examined the effects of extruded linseed and algae *Padina pavonica* extract on the reproductive performance, milk production, and gastrointestinal health of female rabbits. Thirty-six nulliparous New Zealand White female rabbits were randomly assigned to three groups (*n* = 12) with different diets. The control group (CNT) received a standard diet, while the other two groups received modified isoenergetic diets in which part of the CNT diet ingredients were replaced with 5% extruded linseed (L5%) and 5% extruded linseed plus 0.2% *Padina pavonica* algae extract (L5%PP). The rabbits were monitored from artificial insemination until the weaning of the rabbit kits, evaluating different reproductive parameters. Our results indicate that extruded linseed and alga *Padina pavonica* extract did not affect the feed intake or body weight of female rabbits. Additionally, no clinically significant histological changes were observed at the gastrointestinal level. The reproductive parameters, including litter size, litter weight, and milk yield, showed no significant differences among groups. Notably, perinatal and pre-weaning mortalities were reduced in litters born to females receiving omega-3 integrated diets (*p* < 0.05). While these findings are promising, further studies are needed to confirm these results and explore the specific mechanisms by which omega-3 affects reproductive function and litter health.

## 1. Introduction

Rabbit farming relies on frequent breeding, and its success critically depends on the reproductive performance of female rabbits. Nevertheless, the intensive reproductive rhythms employed in commercial rabbit farms often induce a negative energy balance, ultimately reducing female rabbits’ fertility [1]. Primiparous rabbit does are particularly vulnerable to this negative energy balance, as their growth is not fully completed [2]. They also exhibit a low feed intake that fails to compensate for the high-energy demands required for milk production [3]. This energy deficit significantly impacts their body condition and hormonal status, resulting in reduced fertility and, ultimately, a reduced productive lifespan [4]. For these reasons, recent studies in cuniculture prioritize enhancing both productive and reproductive performance and the welfare of this species [5]. To achieve these goals, the scientific community has been investigating the potential benefits of incorporating various nutraceuticals into the rabbits’ diet [5,6,7]. Nutraceuticals are natural chemical substances derived from food sources that provide health benefits in addition to the basic nutritional value found in foods [8]. These natural substances are considered functional foods and have been successfully used in rabbit farming due to their nutritional and therapeutic properties [9,10]. Nutraceuticals can be added to the rabbit diet separately or in combination to improve the productive, reproductive, physiological, and immunological performances, which impact rabbit welfare [9]. Furthermore, based on current research, plant- and animal-based nutraceuticals are recommended as natural and safe feed additives in livestock nutrition and represent important substitutes for artificial drugs, such as antibiotics [9]. This latter aspect is particularly crucial in the rabbit farming sector, where reducing the use of antibiotics and enhancing overall animal welfare is required by the new European Union guidelines. Among different nutraceuticals, plant and animal products rich in omega-3 polyunsaturated fatty acids (n-3 PUFAs) have drawn research interest due to their wide physiological benefits [10,11]. Dietary n-3 PUFAs play a crucial role in various physiological processes in both humans and animals, positively influencing growth [12], reproduction [12], blood lipid and glucose metabolism [13,14], cardiovascular function [15,16], immune response [17], brain development, and vision [18]. Since animals cannot synthesize PUFAs, they must be obtained from the diet [12]. Through a series of reactions, these essential fatty acids serve as precursors of very-long-chain fatty acids (VLC-PUFAs). Specifically, linoleic acid (LA, C18:2 n-6) is the precursor of arachidonic acid (ARA, C20:4 n-6), while α-linolenic acid (ALA, C18:3 n-3) is the precursor of eicosapentaenoic (EPA, C20:5 n-3), docosapentaenoic (DPA, C22:5 n-3), and docosahexaenoic (DHA, C22:6 n-3) acids, which are major constituents of membrane phospholipids in nervous and reproductive tissues, and play crucial roles in maintaining health [18,19]. Essential fatty acids also serve as precursors of eicosanoids, such as prostaglandins, leukotrienes, and thromboxanes [20], with distinct metabolic and functional properties. Notably, n-6 PUFA eicosanoids tend to be prothrombotic and pro-aggregatory, while n-3 derivatives exhibit anti-inflammatory, anti-proliferative, and anti-atherosclerotic activities [21,22]. Both human and animal diets typically exhibit a higher n-6-to-n-3 PUFA ratio, potentially limiting the conversion of beneficial n-3 PUFAs. Maintaining a balanced n-6:n-3 PUFA ratio, ideally ranging from 3:1 to 1:1, is therefore crucial for promoting health.

In rabbit farming, dietary n-3 PUFAs can be provided through plant-based sources rich in ALA, such as linseed, or through sources rich in LCP-PUFAs, such as fish oil and marine algae, which are abundant in EPA, DPA, and DHA [19,22]. While extensive research has examined the effects of linseed and fish oil in rabbit diets, marine algae are only recently emerging as sustainable and beneficial feed sources rich in a variety of bioactive compounds, including PUFAs (particularly DHA), polysaccharides, proteins, polyphenols, vitamins, and minerals [23]. The growing interest in algae for animal nutrition is driven by the need to identify new bioactive substances that enhance animal health and improve the sustainability of animal production [24]. Although numerous studies have examined the potential benefits of different algae, the chemical composition and bioactive compounds content of marine algae can vary due to factors such as algal species, harvesting season, environmental conditions, and geographic location [24]. Despite the increasing body of research on algae, *Padina pavonica* remains unexplored, with no published studies on its use as a rabbit feed supplement or its impact on female fertility in any animal species.

Our hypothesis is that the dietary inclusion of extruded linseed can positively impact the female rabbits’ reproductive performance and that the addition of *Padina pavonica* algae extract, as a further source of n-3 and other bioactive compounds, can amplify the beneficial effect of linseed. Therefore, this study aims to assess the effects of dietary inclusion of 5% extruded linseed and 5% extruded linseed plus 0.2% *Padina pavonica* algae extract on the reproductive performance, milk production, and the gastrointestinal histology of nulliparous female rabbits.

## 2. Materials and Methods

### 2.1. Animals and Experimental Design

The experimental trial was conducted in Azienda Agricola Brachino Patrizia, a commercial rabbit farm located in Central Italy. This research was part of the PRIMA project “Omega Rabbit: food for health BenefIT”, funded by the European Union. The experimental protocol was approved by the Ethical Committee of the Department of Veterinary Medicine of the University of Milano (OPBA_18_2021). The animals involved in the experimental trial were reared in accordance with Legislative Decree No. 146, implementing Directive 98/58/EC, which outlines the minimum standards for the protection of animals kept for farming purposes. Every possible measure was taken to minimize animal distress and to employ the minimal number of animals necessary to ensure the reliability of the results. In addition, to ensure the animals’ well-being, the designated farm veterinarian conducted regular health inspections.

Thirty-six nulliparous New Zealand White female rabbits, aged 4 months, were individually housed in conventional cages (L × W × H:75 × 38 × 25 cm) under controlled environmental conditions, with the temperature maintained between +18 °C and +23 °C, relative humidity ranging between 60% and 75%, and a lighting schedule set at 16 h of light and 8 h of darkness. The female rabbits were randomly assigned to three experimental groups (n = 12/group) according to different pelleted diets, formulated to meet the nutritional requirements of the animals [25]. The control group (CNT) was fed a standard diet, while the other two groups received modified diets in which part of the ingredients of the CNT diet were replaced with 5% extruded linseed (L5%) and 5% extruded linseed plus 0.2% *Padina pavonica* algae extract (L5%PP), respectively (details are shown in Table 1 and in the following paragraph). To ensure a balanced diet for each experimental group, the incorporation of *Padina pavonica* algae extract in the L5%PP group was limited to 0.2%. This decision was based on the high ash content of the extract and the literature reporting negative effects on palatability from higher levels of algae inclusion [26,27]. Furthermore, the algae cannot be used in large quantities due to its unappealing taste, and small amounts do not appear to be sufficient to significantly boost the PUFA intake [28].

The rabbit does were fed an increasing daily ratio of feed, starting with 130 g per day, then the dose was progressively increased according to the physiological needs of pregnancy and lactation [25]. After the artificial insemination, the daily feed amount was increased by 30 g per week throughout pregnancy. During lactation, adjustments were made based on litter size, with a maximum average of 450 g per day. Fresh water was always available ad libitum.

Following a 60-day nutritional adaptation period, the rabbit does were artificially inseminated once they reached 6 months of age. Artificial insemination (AI) was performed using sterile insemination pipettes containing 0.2 mL of diluted (1:5) fresh heterospermic semen. Ovulation was induced by an intramuscular injection of 10 μg of synthetic gonadotropin-releasing hormone (GnRH; Receptal, Hoechst-Roussel Vet, Milan, Italy) immediately prior to AI. At the time of AI, receptivity was determined by assessing and recording the vulvar color. Twelve days after AI, pregnancy was diagnosed through abdominal palpation.

After parturition, each doe was kept with her own litter, and no adoptions of kits between does were carried out. The lactation of the female rabbits was controlled by opening the nest once a day, and the milk production of the does was evaluated from the time of parturition until the kits reached 18 days of age. In order to evaluate the milk production, the does were weighed before and after milking. The rabbit kits were weaned at 35 days of age, after which they were separated from their mothers.

During the experimental period, the does feed intake was registered daily (calculated by subtracting the amount of refused feed from the total amount provided each day, at the same time), while body weight (BW) was recorded weekly from the day of AI until day 21 post-partum using an electronic scale (Isolad–Vignoli–Forlì, Forlì, Italy). Furthermore, the following reproductive parameters were evaluated: receptivity (based on the color of the vulva, classified as white, pink, red, and purple) [6], fertility (calculated as the number of parturitions/number of inseminations × 100) [10], litter size at birth, litter size at weaning, litter weight at birth, litter weight at weaning, and perinatal and preweaning mortality. The perinatal period comprised the initial 48 h following birth [3], while the pre-weaning mortality was calculated as the percentage of weaned kits/litter following the perinatal period [5].

At the end of the experimental trial, the animals were slaughtered in an authorized slaughterhouse, following the European Union regulations, specifically Council Regulation No 1099/2009 governing the protection of animals at the time of slaughter. Immediately after death, the gastrointestinal tract of the animals was carefully removed, and distinct samples from different segments (including stomach, duodenum, jejunum, ileum, caecum, and colon) were individually collected in sterile 15 mL tubes. Specifically, stomach samples were collected at the level of the body on the greater curvature; duodenal samples were taken 10 cm distally to the pylorus; jejunal samples were obtained from the midsection, approximately 70 cm distal to the pylorus; ileal samples were collected 10 cm from the sacculus rotundus; cecal samples were taken 10 cm proximal to the sacculus rotundus; and colonic samples were collected from the proximal ascending colon, 10 cm from the ampulla coli. These samples were then preserved in 10% formalin until undergoing histological examination.

### 2.2. Diets

The experimental pelleted diets provided to the rabbit does were isoenergetic and formulated in accordance with dietary recommendations for female breeding rabbits [25] (Table 1). The proximate chemical composition of the diets was determined following the AOAC methods [29], while the analysis of the fiber fraction (NDF, ADF, and ADL) was performed according to Van Soest and Robertson [30] (Table 2).

The raw materials presented a different proximate chemical composition, reported in Table 3. In particular, extruded linseed was predominantly composed of lipid fraction (44.94%), whereas algae extract contained a significant proportion of ash (77.86%).

Table 4 presents the fatty acid profile of both raw materials (extruded linseed and algae *Padina pavonica* extract) and experimental diets, which were analyzed according to Foch et al. [32] using GC-FID (Varian 4500). Fatty acid methyl esters (FAME) were used to identify the different fatty acids. In detail, total lipids were extracted from 10 g of sample and esterified according to Christie [33]. One mL of each solution containing fatty acid esters was transferred into vials for the gas chromatographic analysis. The separation of FAME was performed using a Varian gas chromatograph (CP-3800) equipped with a flame ionization detector (FID) and an Agilent capillary column (100 m × 0.25 mm, CPS Analitica, Milan, Italy) coated with a DB-Wax stationary phase (film thickness of 0.25 µm). Injector and detector temperatures were set at 270 °C and 300 °C, respectively. The carrier gas was helium at a flow rate of 0.6 mL/min. The oven temperature was programmed as follows: from 40 °C (1 min hold) to 163 °C (10 min hold) at 2 °C/min ramp, to 180 °C (7 min hold) at 1.5 °C/min, to 187 °C (2 min hold) at 2 °C/min, and to 230 °C (25 min hold) at 3 °C/min. Single fatty acid methyl esters were identified by comparing their retention time with the retention time of commercially available FAME standard mixture (FAME mix Supelco 2560, Sigma-Aldrich, Darmstadt, Germany). C21:0 methyl ester (CAS number 2363-71-5; Merck H5149, Darmstadt, Germany), eluted under the same conditions of the samples, was used as internal standard (1 mg/100 µL of added solution).

The fatty acid profile of the raw materials revealed a greater prevalence of saturated fatty acids (SFAs) in the algae *Padina pavonica* extract, primarily attributed to palmitic (C16:0) and oleic (C18:1) acids. Conversely, extruded linseed exhibited a higher concentration of α-linolenic acid (ALA), constituting 52.19% of the total fatty acid composition.

In terms of the fatty acid profiles of the experimental diets, both L5% and L5%PP diets exhibited higher concentrations of n-3 fatty acids compared to the CNT diet (Table 4). Notably, the main PUFA fraction in the CNT diet was n-6, primarily composed of linoleic acid (LA).

Despite containing 0.2% algae *Padina pavonica* extract, the L5%PP diet exhibited minimal differences in fatty acid composition compared to the L5% diet. These limited differences were primarily observed in the presence of long-chain fatty acids (0.13% EPA, 0.05% DPA, and 0.06% DHA).

### 2.3. Histological Analysis

The histological examination was performed on different segments of the gastrointestinal tract, including the stomach, duodenum, jejunum, ileum, cecum, and colon.

All the samples were first fixed in 10% neutral buffered formalin, then dehydrated, and finally embedded in paraffin following standard protocols. Four-micrometer tissue sections were routinary stained with H&E and examined for assessment with an optic microscope (Olympus BX50) (Olympus Italia s.r.l, Segrate, Italy). Histological analysis evaluated leukocyte infiltration in the lamina propria, with infiltration graded on a scale from 0 (absent) to 3 (marked) based on the percentage of section, specifically, score 0 (no tissue affected), score 1 (<20% of section), score 2 (21–60% of section), and score 3 (>61% of section). The same criteria were applied to assess mucosal degeneration, necrosis, and hyperplasia. The mean value of the villi height (from the apex to the base) and thickness (from one lateral extremity to the other) in the duodenum, jejunum, and ileum were measured by analyzing 5 villi at 10× magnification (×100). Similarly, colon and cecum thickness was measured from the luminal aspect of the lamina propria to the muscularis mucosae in 5 fields at 40× magnification (×400). A photographic representation of the measurement method can be found in the Appendix A.

### 2.4. Statistical Analysis

Diagnostic graphs and Kolmogorov–Smirnov tests were used to check assumptions and identify outliers. Eight milk production values were eliminated because they were considered outliers. Raw data are presented in the figures.

When severe abnormal distributions were found, the data were analyzed with nonparametric tests. In particular, the Kruskal–Wallis test was used to compare feed intake among diets for each physiological state (this variable was almost constant), and villi heights and thicknesses (their distribution was not normal and the L5%PP group showed great variability). Bonferroni correction was applied for pairwise comparisons. Results were expressed as medians (Md), first (Q1), and third (Q3) quartiles.

The Linear Mixed model with first-order autoregressive covariance structure was used to analyze BW in each physiological state, milk production, and litter size. These models included animals and days after AI or after parturition as subjects and repeated factors, respectively. The models evaluated the main effects of time (six levels for BW: 0, 7, 14, and 21 days after AI or parturition; 18 levels for milk production which correspond to the first 18 days of lactation), group (three levels: CNT, L5%, and L5%PP), and their interaction. Only group effect was included to analyze weight and litter size. The number of rabbit kits was included in the models as a covariate to analyze differences in milk production and litter weight. The results of the covariate effect were expressed by the b parameter and its standard error (SE). Sidak adjustment was used for conducting multiple comparisons.

Receptivity (only two outcomes were found: red and pink), fertility, perinatal, and pre-weaning mortality were analyzed by generalized linear models. Binomial distribution and logit link function were used to analyze receptivity and fertility, while Poisson distribution and log link function were used for mortality.

The associations between each group and histological scores were evaluated with the eta coefficient [34,35]. The eta is a measure of nominal-by-interval association that ranges from 0 (indicating no association among the variables) to 1 (indicating a high degree of association) [35]. The eta was interpreted as a small association if it was <0.3, medium if 0.3 ≤ eta < 0.5, and large if eta was ≤0.5 [36]. Moreover, eta^2^ (the percent of variance in the dependent variable explained by the group variable [35]) was calculated for large effect sizes. The results of Fisher’s exact and z tests (that compare the column proportions) are also reported.

Statistical analyses were performed with SPSS Statistics version 25 (IBM, SPSS Inc., Chicago, IL, USA). We defined *p* ≤ 0.05 as significant, and 0.05 < *p* < 0.1 as a trend.

## 3. Results

### 3.1. Feed Intake and Body Weight

The groups had no differences in feed intake during pregnancy and lactation (Appendix A).

The female rabbits’ BW was affected only by time (*p* < 0.001). Specifically, during pregnancy, BW progressively increased (*p* < 0.05; Figure 1a), while during lactation, it decreased within one week after parturition (−399 ± 20 g, *p* < 0.001; Figure 1b). The group and time x group interaction was not significant during either pregnancy or lactation.

### 3.2. Reproductive Performance

Figure 2 shows the detailed data of receptivity and fertility. Concerning the rabbits’ sexual receptivity (Figure 2a), the L5% and L5%PP groups demonstrated a higher percentage of does with red vulvas (75.0 ± 12.5%) compared to the CNT group (58.8 ± 14.2%), but the statistical analysis does not reveal significant differences (*p* = 0.598). A comparable outcome was observed regarding the rabbits’ fertility, as both L5% and L5%PP experimental groups exhibited higher pregnancy rates (83.3 ± 10.7%) compared to the CNT group (66.7 ± 13.6%), even though statistical analysis could not find significant differences among groups (*p* = 0.537; Figure 2b).

No differences were found in litter size among the groups, neither at birth nor weaning. However, both the L5% and L5%PP groups showed the lowest perinatal (*p* < 0.001) and pre-weaning (*p* < 0.05) mortality rates (Table 5). Moreover, litter weight was influenced by the litter size (*p* < 0.01) but not by the experimental group.

### 3.3. Milk Production

The rabbits’ milk production (Figure 3) progressively increased until day 18 post-partum (*p* < 0.001) and was influenced by the litter size (b = 5.4 ± 0.9, *p* < 0.001). No differences in milk production were found among the experimental groups.

### 3.4. Histological Examination of the Gastrointestinal Tract

At histological examination, mucosal degeneration, necrosis, and/or hyperplasia were not observed in any segment. Leucocyte infiltration in the lamina propria was a common finding. The eta coefficient indicated a large association with a mild leukocyte infiltration (score 1) with the group in the duodenum (eta = 0.764) and colon (eta = 0.632; Appendix A). In the duodenum, the proportion of the score 1 was greater in group L5% than in groups CNT and L5%PP (*p* < 0.05). Conversely, in the colon, the proportion of the score 1 for leukocyte indicator was greater in CNT than in L5% and L5%PP groups (*p* < 0.05).

No differences between groups were found in any of the measured parameters (villi height and thickness in duodenum, jejunum and ileum, and thickness of the mucosa in colon and cecum) (Appendix A). A visual representation of the findings for the gastrointestinal tract is shown in Figure 4.

## 4. Discussion

There is substantial evidence indicating that an n-3 PUFA dietary inclusion affects female reproduction in different species through various mechanisms [12,22]. Both n-6 and n-3 PUFA reflect the dietary intake [37] and are involved in steroid metabolism [12] and the functionality of sperm and oocytes. Despite the theoretical potential for both positive and negative impacts, there remains limited understanding of the comprehensive effects of PUFAs on fertility [12]. In this context, previous studies have investigated the use of n-3 PUFAs on both male and female rabbit reproductive performances using linseed and fish oil as sources [10,38,39], while no studies have previously assessed the combined effects of extruded linseed and alga *Padina pavonica* extract.

Based on our results, the feed intake of the rabbits did not differ among groups during the experimental trial, suggesting that both extruded linseed and *Padina pavonica* algae extract had no negative effects on the diets’ palatability. This is an interesting finding, as previous research indicated palatability issues with different marine algae in the rabbits’ diets [26,40]. Our study is the first to examine the effects of *Padina pavonica*, incorporated in the diets at 0.2% level due to previous concerns about marine algae’s high ash content and its potential negative impact on palatability, feed intake, and weight gain [26,40].

The feed intake is a critical factor in rabbit farming, especially for primiparous female rabbits in intensive breeding systems. Indeed, primiparous rabbits struggle to meet the elevated nutritional demands of lactation [41] due to their limited feed intake capacity. This, in turn, results in reduced body fat reserves and contributes to lower reproductive performance [4]. While n-3 PUFA dietary supplementation has been explored as a potential solution, previous research [39,40,42,43], in accordance with our findings, suggest it does not significantly impact the feed intake.

Additionally, this study’s results demonstrated no pathological effects in the examined gastrointestinal segments of the treated animals. The only association was reported for a mild leukocyte infiltration in the group treated with extruded linseed. Taking into account the absence of significant morphometric differences in the villi and the absence of a clinical manifestation, its clinical relevance appears negligible. Thus, the histological examination of the gastrointestinal tract in this study revealed positive results, indicating that the diets were well tolerated by the animals and did not adversely affect their gut health. The absence of significant pathological changes or clinical symptoms supports the conclusion that these diets are safe for rabbit’s consumption. This finding is consistent with previous research, which has shown that diets enriched with n-3 PUFAs, such as those containing linseed and fish oil, do not adversely affect the gastrointestinal health of rabbits [44]. Additionally, studies on fattening rabbits supplemented with EPA and DHA from fish oil found no differences in crypt depth and villi length, both of which remained within normal dimensions, further indicating the safety and tolerability of these dietary components [45,46]. The combined effect of linseed and *Padina pavonica* algae extract on gastrointestinal health is particularly noteworthy, as it represents an innovative finding previously unexplored in research.

In this study, the rabbit does’ sexual receptivity was determined by assessing the color of the vulva at the time of artificial insemination, as commonly used in rabbit farming. Only rabbits exhibiting red vulvas were considered as receptive. The color of the vulva in female rabbits represents an external sign of estrus and is highly indicative of receptivity during artificial insemination, serving as a reliable predictor of the estrus cycle phase, as it becomes reddish under the influence of estrogens [47]. In our study, the groups supplemented with extruded linseed and the combination of extruded linseed plus the *Padina pavonica* algae extract demonstrated a higher percentage of does with red vulvas compared to the control group. Although the sample size was insufficient to achieve statistical significance, this result is noteworthy and merits further investigation in future studies. In general, a dietary integration with n-3 PUFAs may enhance the sexual receptivity of female rabbits, as shown in previous studies utilizing both linseed and fish oil [22,48]. This increase in receptivity may be attributed to hormonal modifications, particularly concerning estrogen plasma concentrations [26,39,43,48]. In line with this, previous studies on rabbit does have consistently demonstrated elevated plasma estrogen concentrations following the administration of linseed and fish oil as the source of n-3 PUFAs in the diet [26,48,49]. Furthermore, female rabbits have the capability to store dietary n-3 PUFAs in the periovarian adipose tissue, serving as an optimal reservoir of readily available long-chain PUFA (such as EPA, DPA, and DHA) [22]. These stored fatty acids can play a crucial role in enhancing various physiological and metabolic processes within the ovaries, ultimately leading to improvements in ovarian function and follicular environment [22,38,39]. To fully elucidate these mechanisms, future studies incorporating hormonal analyses are essential.

Regarding the rabbits’ fertility, both supplemented experimental groups exhibited higher pregnancy rates compared to the control group; however, statistical analysis could not find significant differences among groups. Although a larger sample size is needed to validate this finding, research on rabbits has shown improvements in fertility following the dietary administration of n-3 PUFAs. As a matter of fact, dietary n-3 PUFAs can increase progesterone blood levels during the pre- and post-implantation phases of gestation [43], thereby enhancing implantation, placentation, and fetal survival [50]. Efficient implantation and placentation processes are crucial for higher post-implantation survival rates of fetuses, ultimately leading to improved fertility and, consequently, better reproductive efficiency in female rabbits. In addition to higher progesterone concentrations, several studies suggest this improvement might also be linked to a reduction in the production of 2-series prostaglandin derivatives from n-6 PUFAs [51,52]. In particular, the 1- and 2-series prostaglandins (PG) are derived from the n-6 PUFA, dihomo-γ-linolenic acid (DGLA; C20:3n-6), and arachidonic acid (AA, C20:4n-6), respectively, whereas the 3-series PG are derived from eicosapentaenoic acid (EPA) [53]. Several studies suggest that dietary n-3 PUFAs can reduce the secretion of 2-series PG by the endometrium, thereby potentially preventing early embryonic death [51,54]. Prostaglandins PGF_2α_ and PGE_2_ play crucial roles in controlling female reproductive processes, including follicular development, ovulation, corpus luteum lifespan, pregnancy, and parturition [55]. Specifically, reducing PGE_2_, the most pro-inflammatory prostaglandin, can decrease inflammation associated with ovulation, while inhibiting endometrial PGF_2α_ results in an anti-luteolytic effect that enhances embryo survival [12,51]. Specifically, a reduced uterine secretion of PGF_2α_ during early embryonic development could prevent the initiation of luteolysis, thus promoting the establishment of pregnancy [56]. This allows the conceptus more time to grow before potential luteal regression occurs [51]. Moreover, n-3 PUFAs can directly influence the different phases of gestation by facilitating placental blood flow to the fetuses, consequently improving fetal development and growth [50]. This is because n-3 PUFAs increase the ratio between prostacyclin (a vasodilator) and thromboxane (a vasoconstrictor). This promotes vasodilation and reduces blood viscosity, ultimately leading to an improved blood flow to the fetus. Supporting this concept, Rodríguez et al. [43] found no change in the percentage of viable fetuses, but observed greater fetoplacental development in terms of fetus size and placental efficiency in rabbits fed diets rich in EPA and DHA supplemented from fish oil. This resulted in larger and heavier newborns, as reported in other studies [48,49]. On the other hand, in our study, no significant differences in litter size and weight were observed among groups, neither at birth nor weaning, which is consistent with previous studies [39] that found no significant effects of n-3 PUFA dietary integration on both litter size and weight at birth and weaning.

Interestingly, our study revealed that both supplemented groups showed significantly lower perinatal and pre-weaning mortality rates. This finding might be explained by the fact that dietary n-3 PUFAs during pregnancy promote early neuronal development and regulate several neurochemical aspects related to stress response, growth, and cognitive functions, all of which are very important for the vitality of newborns, especially in the first days of life [57]. Fatty acids can be transferred to the fetus via the placenta, either directly from the maternal diet or through the lipolysis of stored reserves [53]. The efficiency of this transfer depends on factors like the timing, duration, amount of fatty acid administered, and specific type of fatty acid provided. Notably, the rabbit placenta has a hemochorial structure, which optimizes fetal–maternal exchange compared to other species [58]. This efficient exchange system could explain why dietary n-3 PUFA during pregnancy might contribute to reduced kit mortality rates in the neonatal period.

In addition, newborn mammals can also benefit from n-3 PUFAs through milk consumption, as the fatty acid composition of milk typically reflects the composition of the mother’s diet [57]. Studies suggest that supplementing the rabbit does’ diet with n-3 PUFA could alter their milk composition and enhance the litter’s defense mechanisms [59]. Indeed, several studies have demonstrated the manipulation of the fatty acid profile of rabbit does’ milk through dietary interventions, resulting in an increase in n-3 PUFAs such as ALA or EPA, DPA, and DHA with the supplementation of linseed or fish oil, respectively [22,43]. One of the mechanisms by which n-3 PUFAs could potentially influence neonatal survival is by enhancing their immune system. Immunoglobulin G (IgG) found in colostrum serves as the primary source of antibodies, which boost the passive immune system of neonates. Studies have shown that colostral IgG concentrations are higher in sows that are fed an n-3 PUFA enriched diet [60]. Furthermore, research has shown that rabbit kits weaned from does that are fed a diet enriched with n-3 PUFAs, and who continued consuming the same diet after weaning, experienced reduced mortality rates in a farm affected by epizootic rabbit enteropathy [61]. In this context, future research could consider examining the modifications in the fatty acid profile of does’ milk and investigate how these modifications affect the transfer of beneficial fatty acids to the offspring and their potential impact on health. It is worth noting that milk analysis was not conducted in this study due to the challenges associated with sampling.

## 5. Conclusions

Our results suggest that the dietary incorporation of extruded linseed and alga *Padina pavonica* extract as a source of n-3 PUFAs in nulliparous rabbit does (1) did not significantly affect feed intake or body weight, suggesting no negative effects on the diets’ palatability; (2) did not induce any pathological effect at the level of the gastrointestinal tract; (3) showed a tendency to enhance receptivity and fertility, although a larger sample size is needed to statistically confirm this trend; and (4) reduced perinatal and pre-weaning mortality rates in rabbit kits. Overall, these results suggest that both extruded linseed and alga *Padina pavonica* extract can be safely incorporated into rabbit diets. Furthermore, according to these findings, a dietary n-3 PUFAs inclusion appears to be a promising strategy to physiologically improve the productive and reproductive performance of nulliparous female rabbits and the litter survival. However, our results do not support the hypothesized improving effect from adding the alga to linseed. Nonetheless, no difference was found when the *Padina pavonica* algae extract was added to linseed, demonstrating the absence of negative effects of the alga. Moreover, given the absence of adverse effects at 0.2%, future studies could explore higher inclusion levels of *Padina pavonica* algae in the rabbit diet. The validation of these results through studies involving larger sample sizes and further exploration of the underlying mechanisms of action are necessary.

## Figures and Tables

**Figure 1 vetsci-11-00457-f001:**
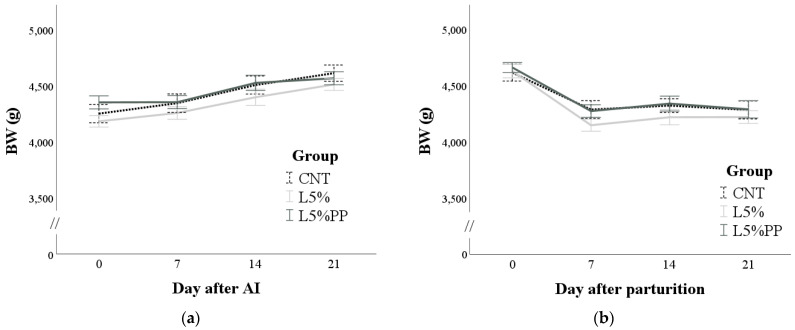
(**a**): Changes in the body weight (BW) of the rabbit does of the experimental groups during pregnancy from the day of artificial insemination. (**b**): Changes in the body weight (BW) of the rabbit does of the experimental groups during lactation from parturition until day 21 post-partum. CNT group: control group receiving a standard diet; L5% group: receiving a diet modified with 5% extruded linseed; L5%PP group: receiving a diet modified with 5% linseed and 0.2% algae *Padina pavonica* extract.

**Figure 2 vetsci-11-00457-f002:**
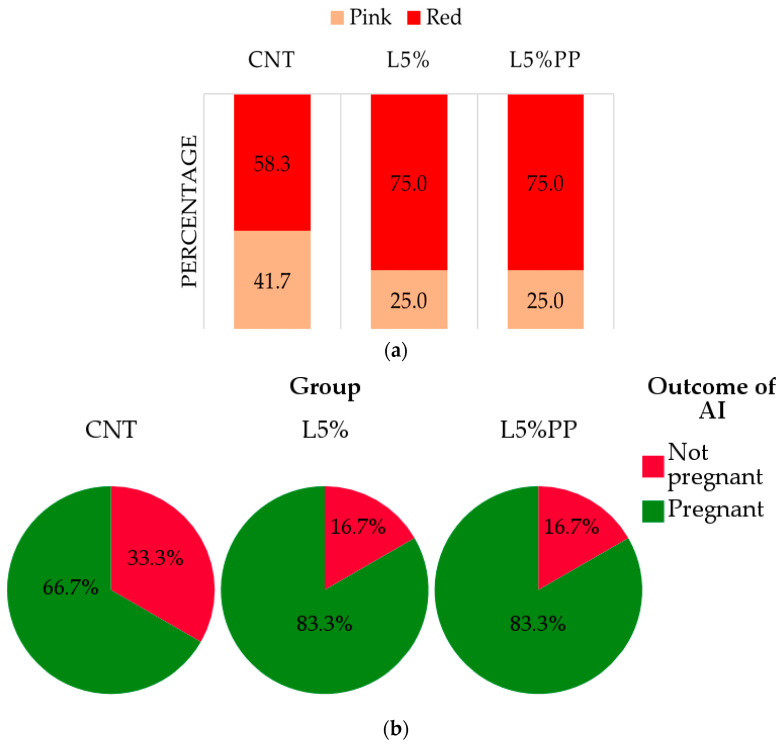
Receptivity based on the color of the vulva classified as red or pink (panel (**a**) and fertility (panel (**b**) of rabbit does of the experimental groups (CNT group: control group receiving a standard diet; L5% group: receiving a diet modified with 5% extruded linseed; L5%PP group: receiving a diet modified with 5% linseed and 0.2% algae *Padina pavonica* extract).

**Figure 3 vetsci-11-00457-f003:**
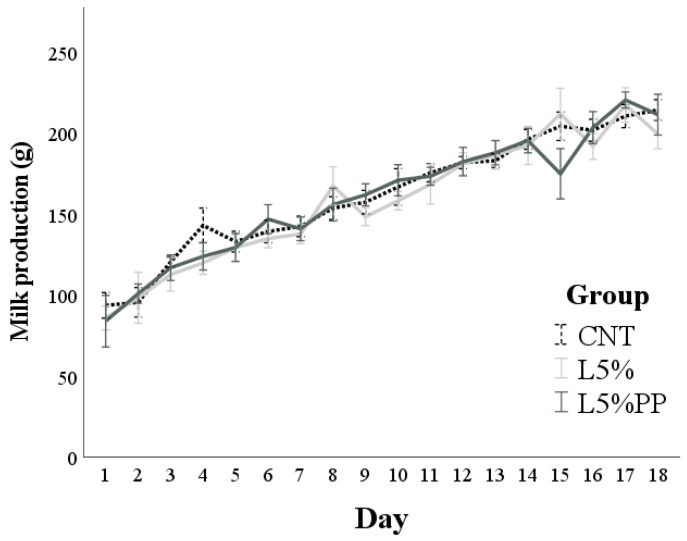
Milk production from day 1 to day 18 post-partum of the experimental groups (CNT group: control group receiving a standard diet; L5% group: receiving a diet modified with 5% extruded linseed; L5%PP group: receiving a diet modified with 5% linseed and 0.2% algae *Padina pavonica* extract). Values are means and standard errors.

**Figure 4 vetsci-11-00457-f004:**
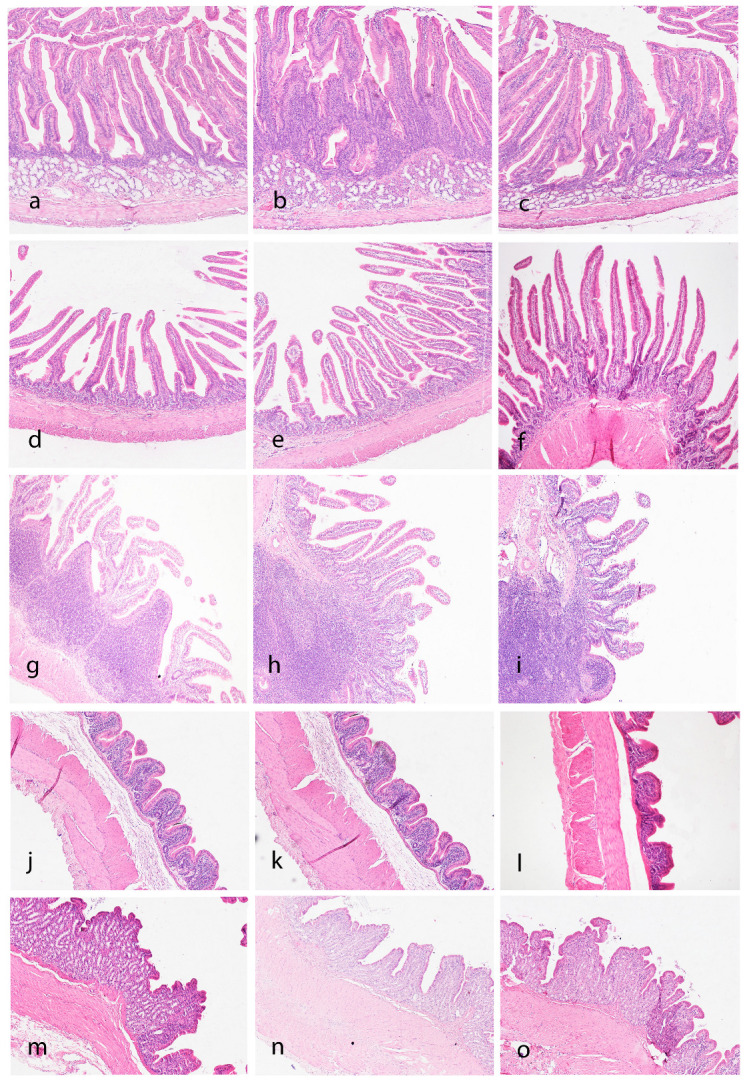
Histologic (pathology and morphometry) assessment of the gastrointestinal tract. Duodenum: CNT (**a**), L5% (**b**), and L5%PP (**c**) groups showing similar morphological feature of the villi. A mild increase in cellularity is seen in the group L5% (**b**). Jejunum: CNT (**d**), L5% (**e**), and L5%PP (**f**) groups showing similar morphological feature of the villi. Pathological changes are not present. Ileum: CNT (**g**), L5% (**h**), and L5%PP (**i**) groups showing similar morphological appearance of the villi over a prominent lymphoid aggregate (Payer’s patch). Cecum: CNT (**j**), L5% (**k**) and L5%PP (**l**) groups showing similar morphological features with mild inflammatory infiltrates in all groups. Colon: CNT (**m**), L5% (**n**), and L5%PP (**o**) groups with similar morphological features with minimal inflammatory infiltrates.

**Table 1 vetsci-11-00457-t001:** Feed formulation (g/kg) of the diets of the three experimental groups: CNT: standard diet; L5%: diet modified with 5% extruded linseed; L5%PP: diet modified with 5% extruded linseed and 0.2% alga *Padina pavonica* extract.

Ingredients (g/kg)	Diets		
	CNT	L5%	L5%PP
Wheat bran	250.7	248.7	249.1
Barley	133.3	130	130
Sunflower seed meal	120	116.7	115
Alfalfa	108.3	130	130
Sunflower husks	100	100	100
Beet pulp	75	56.6	55
Extruded linseed	-	50	50
Full-fat soybean	50	29.5	31
Wheat straw	41.7	20	20
Sugarcane molasses	30	30	30
Wheat	25	25	25
Grape seeds meal	23.3	16.7	16.7
Soya hulls	0	1.67	1.67
Calcium carbonate	16	14.8	14.2
Soybean oil	7.8	-	-
Sodium chloride	4	4	4
Palm oil	3.3	-	-
Carboxymethylcellulose	3	3	3
Oligo vitamin supplement ^1^	2.5	2.5	2.5
Algae “*Padina pavonica*” extract	-	-	2
Lysine HCl	1.6	1.7	1.7
Liquid acidifier ^2^	1.5	1.5	1.5
Magnesium oxide	1	1	1
Methionine hydroxyanalog	0.6	0.7	0.7
Liquid choline	0.5	0.5	0.5
Vitamin E 50%	0.3	0.3	0.3
L Threonine	0.3	0.1	0.1
DL Methionine	0.3	-	-

^1^ Vitamin mineral premix composition: Vitamin A 6,000,000 IU, D3 600,000 IU, E 20,000 IU, K3 1200 mg, B1 800 mg, B2 1600 mg, B6 800 mg, B12 6.0 mg, Biotin 60.0 mg, Niacinamide 16,000 mg, Folic acid 400 mg, Calcium pantothenate 6666 mg. ^2^ Liquid acidifier composition: Formic acid 75%.

**Table 2 vetsci-11-00457-t002:** Chemical composition (g/kg) of the diets of the three experimental groups: CNT: standard diet; L5%: diet modified with 5% extruded linseed; L5%PP: diet modified with 5% extruded linseed and 0.2% alga *Padina pavonica* extract.

Chemical Composition (g/kg) of Dry Matter	Diets		
	CNT	L5%	L5%PP
Dry matter	893.4	894.9	899.4
Crude protein	177.2	183.3	185.9
Ether extract	39.5	62.1	52.2
Ash	79.9	76.9	82.1
NDF ^1^	432.4	396.5	424
ADF ^2^	279	257.2	255.3
ADL ^3^	77.6	73.8	72.2
DE ^4^	9.8324	9.8324	9.8324

^1^ NDF: neutral detergent fiber (%); ^2^ ADF: acid detergent fiber (%); ^3^ ADL: acid detergent lignin (%). ^4^ DE: digestible energy (MJ/kg), estimated according to Maertens et al. [31].

**Table 3 vetsci-11-00457-t003:** Proximate chemical composition (g/kg) of raw materials: extruded linseed and algae *Padina pavonica* extract.

Chemical Composition (g/kg)	Extruded Linseed	Algae *Padina pavonica* Extract
Dry matter	954.6	996.6
Crude Protein	227.6	2.5
Ether extract	449.4	1.6
Ash	31.1	778.6
NDF ^1^	132.4	28.3
ADF ^2^	108.1	19.9
ADL ^3^	40.6	5.6

^1^ NDF: neutral detergent fiber; ^2^ ADF: acid detergent fiber; ^3^ ADL: acid detergent lignin.

**Table 4 vetsci-11-00457-t004:** Fatty acids profile (g/100 g of fatty acids) of raw materials (extruded linseed and algae *Padina pavonica* extract) and experimental diets: CNT: standard diet; L5%: diet modified with 5% extruded linseed; L5%PP: diet modified with 5% extruded linseed and 0.2% alga *Padina pavonica* extract.

Fatty Acid (g/100 g of Fatty Acids)	Raw Material		Diet		
	Extruded Linseed	Algae *Padina pavonica* Extract	CNT	L5%	L5%PP
C12:0	0.00	0.13	0.00	0.00	0.00
C14:0	0.05	1.62	0.18	0.20	0.20
C15:0	0.02	0.00	0.09	0.08	0.08
C16:0	5.97	27.95	13.92	11.14	11.65
C16:1cis9	0.08	1.51	0.18	0.20	0.18
C17:0	0.06	0.00	0.12	0.11	0.10
C17:1	0.03	0.00	0.05	0.05	0.05
C18:0	4.54	12.52	3.07	3.82	3.48
C18:1	20.93	25.84	25.39	23.89	23.98
C18:2cis n-6, LA ^1^	15.11	9.72	47.63	32.41	33.53
C20:0	0.15	0.00	0.28	0.22	0.22
C18:3 n-6, γ-ALA ^2^	0.01	0.00	0.21	0.12	0.19
C18:3 n-3, α-ALA ^2^	52.19	8.98	6.53	23.25	22.63
C20:2	0.00	0.00	0.00	0.00	0.04
C20:3n-3	0.00	0.00	0.00	0.00	0.02
C22:0	0.00	0.00	0.00	0.00	0.23
C22:1	0.00	0.00	0.00	0.00	0.04
C20:4n-6, AA ^3^	0.00	0.34	0.00	0.00	0.08
C22:2	0.00	0.00	0.00	0.00	0.01
C20:5n-3, EPA ^4^	0.00	0.03	0.00	0.00	0.13
C24:0	0.00	0.00	0.00	0.00	0.00
C24:1	0.00	0.00	0.00	0.00	0.03
C22:5n-3, DPA ^5^	0.00	0.00	0.00	0.00	0.05
C22:6n-3, DHA ^6^	0.00	7.63	0.00	0.00	0.06
Tot	99.16	96.25	97.64	95.50	97.00
Others	0.84	3.75	2.36	4.50	3.00

^1^ LA: linoleic acid; ^2^ ALA: α-linolenic acid; ^3^ AA: arachidonic acid; ^4^ EPA: eicosapentaenoic acid; ^5^ DPA: docosapentaenoic acid; ^6^ DHA: docosahexaenoic acid.

**Table 5 vetsci-11-00457-t005:** Reproductive parameters of CNT group: control group receiving a standard diet; L5% group: receiving a diet modified with 5% extruded linseed; L5%PP group: receiving a diet modified with 5% linseed and 0.2% algae *Padina pavonica* extract. Data are mean and standard errors.

Parameter	Group			*p* Value
	CTN	L5%	L5%PP	
Litter size at birth (total n)	6.78 ± 0.70	7.67 ± 0.40	7.29 ± 0.62	0.576
Litter size at weaning (n)	5.89 ± 0.75	6.67 ± 0.41	6.79 ± 0.57	0.507
Litter weight at birth (g) *^†^	427 ± 25	440 ± 20	468 ± 19	0.394
Litter weight at weaning (g) *^†^	4987 ± 199	4814 ± 188	4965 ± 177	0.784
Perinatal mortality (%) *	6.67 c ± 0.86	2.78 b ± 0.56	0.00 a ± 0.00	<0.001
Pre-weaning mortality (%) *	5.56 c ± 0.79	0.00 a ± 0.00	3.33 b ± 0.53	0.016
Milk production (g/d) *	161.9 ± 2.8	158.6 ± 2.6	159.90 ± 2.5	0.679

* Estimated; ^†^ Litter size was included as covariate in the models; Values followed by the same letter in each row do not differ significantly (*p* < 0.05).

## Data Availability

The original contributions presented in the study are included in the article/Appendix A.

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
