# Peer review of "The Effect of Dietary Plant-Derived Omega 3 Fatty Acids on the Reproductive Performance and Gastrointestinal Health of Female Rabbits"

_vetsci, 2024, doi:10.3390/vetsci11100457_

Round 1
Reviewer 1 Report
Comments and Suggestions for Authors
Effect of dietary plant‐derived omega 3 fatty acids on the reproductive performance and gastrointestinal health of female rabbits
General comments
The study was designed to investigate the effects of extruded linseed and algae extract on the reproductive performance and gastrointestinal health of nulliparous rabbit does. The manuscript is well organized and written for the most part. There are some aspects of the methodology that need clarification, especially related to feed formulation and processing. Feed composition was expressed in percentage, but this is not the right (recommended) way to do it. I question whether the inclusion of 0.2% of algae extract (very low level) was effective to change nutrient availability in any way. The results seem consistent, and the conclusions are sound.
Specific comments
Abstract:
Line 59 (and line 147 of the Introduction): I believe that “productive performance” is not the best way to describe one of the objectives of the study, but the gastrointestinal health or histology.
Lines 61-63: You mentioned that the control group received a “commercial” diet. Here is where my questions begin: What does “commercial diet” mean to you? To me a commercial diet is available in the market, formulated and processed by the feed industry. Since a detailed composition of the experimental diets was presented in Table 1, I came to a conclusion that you have formulated, prepared and processed the experimental diets. Please clarify this crucial point. I will get back to it in the Material and Methods section.
Introduction
The introduction is rather long, could be more compact. But if there is no space limitation, it can remain this way.
Material and methods
Sample size (36 females, 12 per group) was relatively small.
Line 169: here you describe the control diet as a “standard diet”. OK.
Lines 169-178: I understood your concerns, but it is unfortunate that there was only one very low level of inclusion of the algae extract. If you had a treatment with a higher level of inclusion, you would come to more reliable results and conclusions.
Line 180: You have increased the amount of feed offered according to pregnancy and lactation needs. Please give more details on how this was done. This is critical to understand the feed intake data. You cited an entire book!
Line 182: There was a 60-day adaptation period to the diets. Does this mean that the females were 6 months old at A.I.? Or did the adaptation period begin before the females were individually caged at 4 months of age (line 163)? Please include this information in the text.
Lines 189-193: It is important to indicate whether the females were reinseminated any time before weaning. This information is important because, in commercial rabbit production, the does are, in general, inseminated 10 to 25days after parturition.
Line 194: how did you record daily feed intake?
Table 1:
· As I mentioned before, “commercial diet” does not seem an adequate terminology. Please change to “standard diet” and use “control diet” (not commercial diet) throughout the manuscript text including tables and figures titles. Do not use commercial diet in any part of the manuscript.
· Diet composition and proximate analyses results must be expressed as g/ kg or g kg-1, according to your reference n. 28 (Blas & Wiseman, 2010).
· DE should be expressed in MJ/ kg, not kcal/ kg (Blas & Wiseman, 2010). How was the DE of the diets determined? Was it estimated?
Table 2: ingredient composition must also be expressed as g/ kg or g kg-1, according to your reference n. 28 (Blas & Wiseman, 2010).
Line 235: ingredients and diets were analyzed by Foch et al. (1957) or according to Foch et al. (1954). Please clarify.
Line 237: What is FAME? Please explain and present a reference.
Table 3: Please use g/100 g (or g 100g-1) of fatty acids, not percentage.
Line 260: please indicate the organ portion where sample collection took place in each case (stomach, duodenum, jejunum, ileum, cecum and colon).
Line 280: It is not clear to me why you used non-parametrical analyses (Kruskal Wallis) for feed intake. This is unusual (to say the least) and relates to previous comments asking for more details on how you increased the amount of feed offered during pregnancy and lactation. And how you measured feed intake.
Line 336: Delete “and fertility”.
Line 342: Delete “significant”.
Table 5:
· Only one significant digit for litter size means is unacceptable, since these are means, not medians. Please present at least two. For example, litter size 6.8 and 7.3 are both rounded to 7!
· For milk production, I believe the units are “g/d”, not simply “g”.
Line 371: “in the height or thickness of “? Please be specific and indicate the exact names of these parameters (villi height, villi thickness).
Comments on the Quality of English Language
English language is good.
Reviewer 2 Report
Comments and Suggestions for Authors
GENERAL COMMENT:
I consider this work is within the scope of “Veterinary Sciences”. It contains information useful in a field in which available information is scarce and of interest for improving feeding of rabbit does to enhance reproductive performance and gastrointestinal health.
However, there are several points to improve. I indicate below these points to be improved in the manuscript.
TITLE:
It is OK.
SIMPLE SUMMARY:
It is OK.
ABSTRACT:
It is OK.
KEYWORDS:
These are OK.
INTRODUCTION:
This section is OK.
MATERIALS AND METHODS:
There are several points to be improved:
Line 164: Please add the dimensions of the cages used to house the does.
Line 169: Please add a bibliographic quotation for the nutritional requirements of the does.
Line 191: Following the criteria of the World Rabbit Science Association, it is better writing “kits” rather than “young rabbits”.
Lines 189-199: It is necessary to indicate whether the litter size was homogenized after birth, through adoptions of kits between does (indicating if so, what litter size was equalized), or whether the does were left with the litter size they had given birth to.
Line 217, with respect to Table 1: Please indicate how the value of digestible energy from feed was determined or estimated.
Line 280: Correct typo: “Kruskal-Wallis”, rather than “Krusjal Wallis”.
RESULTS SECTION:
Overall, this section is OK.
Line 316: Correct the units. These are “g” rather than “gr” because the symbol of gram is “g”, according to the International System of Units, SI.
DISCUSSION SECTION:
Overall, this section is OK.
CONCLUSIONS:
This section is OK.
REFERENCES SECTION:
In general terms, this section has a good adjustment to the style and format of the journal for references. However, I recommend reviewing it for removing typos and correct potential flaws. For example:
Journal titles must be abbreviated.
Latin names of the organisms must be typed in italics.
Do not type article titles with all words in capital letters.
Do not type with uppercase letter the first letter of the species in the Latin names of the organisms.
References 32, 58 are incomplete.
Etc.
TABLES:
Table 1, footnote: indicate the amount of each component of the Vitamin Mineral premix.
Table 5: What is the reason behind the fact that the values of the indexes “Litter size at birth” and “Litter size at weaning” do not have decimal digits? Since these are calculated with the births of the rabbits in each batch, it would normally be written with at least one decimal place.
FIGURES:
Figures 1 and 3: correct the units in the Y-axis: These are “g” rather than “gr” because the symbol of gram is “g”, according to the International System of Units, SI.
SUPPLEMENTARY MATERIAL:
It is OK.
